# High Concordance of CT Colonography and Colonoscopy Allows for the Distinguishing and Diagnosing of Intestinal Diseases

**DOI:** 10.3390/life13091906

**Published:** 2023-09-13

**Authors:** Lyubomir Chervenkov, Nikolay Sirakov, Aleksander Georgiev, Dimitrina Miteva, Milena Gulinac, Monika Peshevska-Sekulovska, Metodija Sekulovski, Tsvetelina Velikova

**Affiliations:** 1Department of Diagnostic Imaging, Medical University Plovdiv, 4000 Plovdiv, Bulgaria; lyubo.ch@gmail.com (L.C.); aleksandar.georgiev@mu-plovdiv.bg (A.G.); 2Research Complex for Translational Neuroscience, Medical University of Plovdiv, Bul. Vasil Aprilov 15A, 4002 Plovdiv, Bulgaria; nikolay.sirakov@mu-plovdiv.bg; 3Department of Diagnostic Imaging, Dental Allergology and Physiotherapy, Faculty of Dental Medicine, Medical University Plovdiv, 4000 Plovdiv, Bulgaria; 4Department of Genetics, Faculty of Biology, Sofia University “St. Kliment Ohridski”, 8 Dragan Tzankov Str., 1164 Sofia, Bulgaria; d.georgieva@biofac.uni-sofia.bg; 5Medical Faculty, Sofia University St. Kliment Ohridski, 1 Kozyak Str., 1407 Sofia, Bulgaria; mgulinac@hotmail.com (M.G.); mpesevska93@gmail.com (M.P.-S.); metodija.sekulovski@gmail.com (M.S.); 6Department of General and Clinical Pathology, Medical University of Plovdiv, Bul. Vasil Aprilov 15A, 4000 Plovdiv, Bulgaria; 7Department of Gastroenterology, University Hospital Lozenetz, 1407 Sofia, Bulgaria; 8Department of Anesthesiology and Intensive Care, University Hospital Lozenetz, 1 Kozyak Str., 1407 Sofia, Bulgaria

**Keywords:** computed tomography, CT colonography, fibrocolonoscopy, gastrointestinal diseases, virtual colonoscopy, incomplete colonoscopy, tumor

## Abstract

(1) Although new imaging methods for examining the GIT with high diagnostic capabilities were introduced, the improvement and implementation of safe, efficient, and cost-effective approaches continue, and GIT diseases are still challenging to diagnose; (2) Methods: We aim to show the possibilities of computed tomography (CT) colonography for early diagnosis of colon diseases using a multidetector 32-channel CT scanner after appropriate preparation; (3) Results: After a colonoscopy was performed earlier, 140 patients were examined with CT colonography. Complete colonoscopy was performed in 80 patients (57.1%) out of 140 who underwent CT colonography. Incomplete colonoscopy was observed in 52 patients (37.2%); in 5 patients (3.6%), it was contraindicated, and in 3 patients (2.1%), it was not performed because of patients’ refusal. We determined that in cases of complete FCS in 95% of patients, CT colonography established the same clinical diagnosis as FCS. In cases of incomplete, refused, or contraindicated FCS in 32.7% (17 patients), FCS failed to diagnose correctly. The main reasons for incomplete colonoscopy were: intraluminal obturation of tumor nature-17 patients (33%), extraluminal obturation (compression) from a tumor formation-4 patients (8%), stenotic changes of non-tumor nature-11 patients (21%), congenital diseases with changes in the length of the lumen of the intestinal loops-7 patients (13%), and subjective factors (pain, poor preparation, contraindications) in 13 patients (25%); (4) Conclusions: Our results confirmed that CT colonography is a method of choice in cases of negative FCS results accompanied by clinical data for the neoplastic process and in cases of incomplete and contraindicated FCS. Also, the insufflation system we developed optimizes the method by improving the quality of the obtained images and ensuring good patient tolerance.

## 1. Introduction

Since the introduction of the colonoscope in the 1960s, the use of colonoscopy has been constantly developing and evolving. Today it is considered the golden standard for diagnosing various diseases of the GIT (gastrointestinal tract). The most important role of colonoscopy is the reliable diagnosis of CRC (colorectal cancer), polyps, inflammatory diseases, and bleeding [1]. The method can be used not only for diagnostic purposes but parallel to the diagnosis; the physicians performing the procedure can easily perform polypectomy and take samples for histological verification of the pathological sample [1].

Ultrasound diagnostics, computed tomography, capsule endoscopy, and endoscopic methods are widely used. In parallel with this positive process based on new technologies, there are trends to limit the X-ray methods in the diagnostic algorithm of diseases of the GIT and use non-X-ray methods [2]. Most often, endoscopic techniques are opposed to X-ray methods, while doctors often forget that accurate and timely diagnosis based on complementary approaches is essential for the patient. Each diagnostic methodology has a particular priority of application and optimal efficiency in different organs and related diseases.

Although widely used, a colonoscopy has certain limitations that narrow the diagnostic value of the method [3]. The proper diagnosis relies on a complete colonoscopy procedure where the colonoscope must reach the cecal valve. There are patients with various anatomical abnormalities or pathological conditions that increase the length of the colon, such as megacolon, that can result in depletion of the length of the colonoscope, thus leading to an incomplete colonoscopy. During the procedure, the tip of the colonoscope can be moved only to a certain degree, leading to a certain number of blind spots that can hide a small polyp or an early cancer site. Abrahams et al. [4] stated nearly 21% of blind spots during an endoscopic procedure. Most often, it is due to acute curvatures or haustration. Another reason for an incomplete colonoscopy could be the impossibility of proceeding with the examination due to the narrowing of the colon. Even though most of these cases are related to narrowing from the main pathology that can allow taking histological samples, many of them are due to pathologies proximal to the stricture that do not allow the colonoscopy to proceed [1]. We were interested in improving all of these obstacles while performing the method.

Another factor that influences the success rate of the colonoscopy is adequate colon preparation. Poor preparation can lead to residual stool that can mimic pathology or intracolonic fluids that can hide more minor findings [1].

Very often, a colonoscopy is contraindicated to be performed, mostly in cases with Cron’s disease or various inflammatory or necrotic conditions. The risk of perforation and other complications should also be considered even though it is smaller (less than 1%) [5].

The patient’s acceptance of a colonoscopy is not great due to discomfort, pain, and fear of sedation. These rates are even lower in patients who do not have colon disease symptoms but need a screening procedure. Other examination methods can aid the proper diagnosis in incomplete, contraindicated, or refused colonoscopy cases.

For example, double-contrast barium enema, an X-ray method for examining the large intestine, was chosen in the pre-CT/MRI era. It has good sensitivity for diagnosing colon cancer—87%. Still, the method’s sensitivity drops to 21% in detecting neoplastic polyps (polyps larger than 1 cm) [6].

On the contrary, during the last 2 decades, capsule endoscopy has gained a lot of significance as an alternative to colonoscopy. This method of examining the colon uses a small camera the size of a pill, containing a power source, a light source, and a camera with Wi-Fi or Bluetooth capabilities [7] is highly tolerated by patients but has a lot of limitations—a high rate of blind spots and low sensitivity due to its passive movement through the colon. The second generation of capsule endoscopy includes two cameras on both ends of the capsule (PillCam, Medtronic), but the significance of this improvement has yet to be determined.

In search of an alternative to FCS, D. Vining introduced CT colonography in 1994 as a new method of examination of the colon [8]. Modern CT technologies contribute to creating new imaging diagnostic methods for studying the gastrointestinal tract with high informative value and possibilities for 2D and 3D spatial reconstruction. These technologies made CT colonography possible. Specific computer products are being developed to implement the new radiological method and improve their quality. CT colonography can potentially become the method of choice in screening neoplastic diseases and polyps.

The use of CT colonography has risen recently, reaching over 100,000 performed in the UK in 2020. Two meta-analyses have shown that CT colonoscopy is considered as accurate as colonoscopy. The disadvantage of the method is that the sensitivity is lower in small lesions (less than 9 mm). CT colonography is considered a very safe method, with no death cases reported, and only a few complications. One of the possible complications is perforation of the lumen, which is easily identified on a CT exam and is far less common than in colonoscopy. Another concern is the use of radiation, but with recently developed techniques and modern CT scanners, this disadvantage is considered very minimal [9].

CT colonography has many advantages compared to fibrocolonoscopy (FCS), relying on inflating the colon’s lumen with air, thus excluding preliminary examinations caused by strictures or changes in the length of the colon. The nature of CT colonography makes it possible to detect pathological findings not only inside the colon’s lumen but outside as well, and even in distant organs. This makes the possibility of staging the process much easier and more accurate. The sensitivity of the CT colonography is as high as the FCS, about 97%. The examination is not invasive and is highly accepted by patients. It does not require sedation, which is another factor for incomplete colonoscopy. The only disadvantages of the method are the use of X-rays which could limit examination during pregnancy, and the impossibility of taking a sample from the process for histological verification. Nevertheless, the method has proven to be used for screening purposes and in cases with incomplete, refused, and contraindicated colonoscopies.

CT colonography had a very important role during the pandemic, providing the possibility of performing a socially distanced total colon examination. During the pandemic, colorectal cancer screening dropped dramatically, as doctors in most countries were instructed to postpone colonoscopy, if possible, in order to prevent spread of the virus. CT colonography played the role as an alternative of colonoscopy, especially in some urgent cases. The main advantage of CT colonography is that it does not need sedation, thus enabling the patients to be dismissed after the procedure. Also, in this case, no presence or help is needed from other medical specialists or family members to support the patient, thus keeping the needed social distance [10].

The study’s aim was to investigate the possibilities of CT colonography as an imaging method in the early diagnosis of colon diseases. In addition, a remote-controlled automatic air insufflation system was developed, constructed, and put into service. Another aim was to compare the relaxation potential of buscolisine and drotaverine on smooth muscle tissue’s tone and contractile activity. The reasons for refusal and incomplete fibrocolonoscopy were also analyzed, and the place of CT colonography in the diagnostic algorithm of colon diseases was determined. Furthermore, although performed in our country, the outcomes from CT colonography are loosely published and could not be compared to the world’s data.

## 2. Materials and Methods

### 2.1. Subjects

In our study, we present 140 patients examined with CT colonography after a colonoscopy was performed earlier. Sixty-two of the patients were men, and 78 were women. The patients were hospitalized at University Hospital St. Georgi, Plovdiv, from 01.2022 to 05.2023. The average age of men was 57.48 ± 14.61 years and women was 62.66 ± 15.34 years.

The distribution of patients according to the type of antispasmodic administered is presented in Table 1, and their distribution according to the type of preliminary preparation is shown in Table 2.

### 2.2. Methods

Preliminary preparation began 3 days before the examination with a diet that did not include gas-forming and heavy foods. One day before the CT colonography, the patient took X-prep 75 mL, followed by 2–3 L of water. Preliminary relaxation of the intestinal musculature was carried out with Hyoscine butylbromide or drotaverine, injected intravenously in a dose of 20 mg via abocate 20 min before the start of the study. CT colonography was performed in the morning on an empty stomach. Patients were made to lie supine, after which a topogram was performed. A rectal balloon catheter was connected to a manual pump or to an automatic air insufflation system, which we developed. Manual insufflation was continued until the patient experienced discomfort, after which it was stopped. Discomfort gives indirect information about the achieved distension, which is close to optimal. In cases of insufficient distension, the introduction of air was renewed after the pain subsided. Automatic insufflation continued until an intracolumnar pressure of 15 mmHg was reached.

All patients were examined with a multidetector 32-channel CT scanner (Siemens Go Up, Germany). The parameters of CT acquisition are tube voltage 130 kV, quality ref. mAs 54, Eff. mAs 73 with CARE Dose4D dose optimization. Acquisition (mm) 32 × 0.7; pitch 1.5; rot. time (s) 0.80. All exams were performed in a supine position, at full inspiration without contrast medium. Two reconstructions were made: The first was with 1.5 mm slice thickness with 1.5 mm increment, Br40 Kernel, Abdominal window, Narrow FAST Planning Width, and FAST 3D with Matrix Size 512, and the second was with 1.5 mm slice thickness with 1.5 mm increment, Br20 Kernel, Mediastinum window, SAFIRE strength 3, Narrow FAST Planning Width, and FAST 3D with Matrix Size 512. The scans were observed in axial, sagittal, and coronal planes.

The automatic air insufflation system is tailored to the specific requirements of CT colonography. The device is electro-mechanical. It creates, registers, and tracks the instantaneous value of the pressure in the intestine. The system provides a gradual, uniform introduction of air into the intestine employing a pump until the preset pressure is reached, the value of which ensures the achievement of a better image. The system is shown in Figure 1.

### 2.3. Ethical Approval

All participants were informed about the study and signed informed consent. The study design was approved by the Ethical Committee of University Hospital St. Georgi, Plovdiv (Institutional Review Board approval No. 00182/01.06.2022).

### 2.4. Statistical Methods

For the statistical analysis of the data, we used descriptive statistics via SPSS v.21 and Excel.

## 3. Results

Our study presents 140 patients examined with CT colonography after a colonoscopy was performed earlier.

A comparison between manual and automatic air insufflation was performed in 36 of our patients. Twenty of the patients were women and 16 were men. Patients were divided into two groups: the first group (n = 20, mean age 58 years) in which air insufflation was performed by an assistant by manual input and the second group (n = 16, average age 61 years) in whom air was introduced automatically by a system for automatic insufflation of room temperature air developed by us. There was no significant difference between the average age of the patients of the two groups (n = 0.209) and between the male–female ratio (n = 0.288).

Complete colonoscopy was performed in 80 patients (57.1%) out of 140 who underwent CT colonography. Incomplete colonoscopy was observed in 52 patients (37.2%); in 5 patients (3.6%), it was contraindicated, and in 3 patients (2.1%), it was not performed because of patients’ refusal.

When comparing the results in patients with incomplete, contraindicated, or refused FCS, it was determined that in cases of complete FCS in 95% of patients, CT colonography established the same clinical diagnosis as FCS, which showed a high concordance between the two methods (Figure 2). In cases of incomplete, refused, or contraindicated FCS in 32.7% (17 patients), FCS failed to diagnose correctly; it was found after a CT colonography.

We gathered the various reasons for incomplete colonoscopy into several groups (Table 3).

Additionally, we presented some of our patients in the group for which the virtual colonography was essential to establish the primary or other conditions. First, we presented a 73-year-old patient who had abdominal pain and constipation. The patient underwent incomplete FCS because of the stenosis of the lumen of the intestine (Figure 3). In this patient, we found tumor formation in the lumen of the colon, which arose from the intestinal wall and caused obstruction. The formation was with irregular margins and surrounded the intestine like a sleeve.

The same patient in the prone position allowed us to visualize carcinoma of the sigmoid colon that rose to the lumen (Figure 4). The differential diagnosis of the finding was residual intestinal contents, which required a change of the patient’s position. Placing the patient in a prone position did not show dynamics in the topic and characteristics of the lesion, which confirmed it was a tumor.

Figure 5 shows a 67-year-old female with flank pain and transitional spasm. The clinical data suggested intestinal tumors. FCS was performed, and there was evidence of extraluminal formation, compressing the colon. Afterward, CT colonography was performed, on which tumor formation arising from the left ovary was observed.

The following case was an 80-year-old patient who underwent FCS, on which there was evidence of compression of the colon. CT colonography showed tumor formation with irregular outlines and inhomogeneous density arising from the caecum (Figure 6).

The last example of our experience is a 67-year-old male (Figure 7) who underwent incomplete fibrocolonoscopy, which was not completed because of stenosis of the intestinal lumen due to functional transitional spasm. Afterward, CT colonography was performed, which showed carcinoma of the stomach with metastasis in the liver.

## 4. Discussion

The emergence of CT colonography is an attempt to search for a fast, accurate, and non-invasive method for diagnosing colon diseases and an alternative to colonoscopy. Optimizations in CT colonography are the subject of various studies, focusing mainly on two directions: better patient tolerance of the method and better image quality [11]. One of the difficulties of performing quality CT colonography is adequate patient preparation. The presence of liquid and intestinal contents can overlap an intraluminal formation or vice versa—stool can be interpreted as a polyp or a tumor. We aimed to investigate the possibilities of CT colonography as an imaging method in the early diagnosis of colon diseases, showing a high concordance between CT colonography and colonoscopy.

Patient preparation was standard and uniform and began a few days before the examination [12]. Written information on dietary regimen and medication exposure was provided to every patient. X-prep and Fortrans were used for medicinal preparation. X-prep provides a dry lavage with no or negligible fluid retention in the intestinal loops. In patients using Fortrans, we observed a significant amount of liquid in the intestinal loops, which requires the patients to be examined in two basic positions—back and prone. Our opinion is that X-prep is more suitable for preparing patients for CT colonography [12].

Drotaverine and hyoscine cause relaxation effects and inhibit the phasic smooth muscle contractions [13]. The two drugs cause different effects on the smooth muscle in different parts of the large intestine. The strength of drotaverine-induced responses was greater than that of hyoscine-induced responses [13]. Our results show that drotaverine is more suitable than hyoscine for relaxation, providing optimal colon distension.

Colonoscopy, a golden standard for diagnosing colon diseases, is not always complete. There are several absolute and relative contraindications, such as patient refusal, acute abdomen, acute inflammations, allergy to the sedation medicaments, life-threatening conditions such as coma, insult infarction, etc. Other contraindications include complications from previous operations: adhesions, volvulus, coloptosis, and anus praeter [14].

CT colonography is an alternative to colonoscopy with suitable patient acceptance and high informative value. CT colonography contraindications include toxic megacolon, bowel obstruction, acute abdomen, claustrophobia, pregnancy, and metal prosthesis, which can obscure the nearby intestines [15].

Good colon distention is the leading condition for obtaining more and better information in CT colonography. With manual insufflation, air enters the colon in pulses, constantly changing pressure [16]. Peak values occur that irritate the smooth muscle of the intestinal wall, which responds with a series of contractions. They cause a local increase in the tone or increased motility. These processes are established mostly in the rectum and the sigmoid area near the air penetration site. There is an initial distension of the rectal area, onset of pain, and subsequent uneven distension of the other column areas.

Automatic insufflation of room temperature with air with our device allows significantly better distension of the bowel lumen. In addition, patients subjectively experience less discomfort with automatic air insufflation. This is most likely due to the more uniform flow, where spastic reactions are less frequent.

CT colonography is the best method of choice in cases of incomplete colonoscopy. Even in cases of intraluminal obturation, CTC can achieve a whole-scale colon examination, even proximal to the obturation, due to air and maximal distension of the colon wall [17]. This also helps with the disease’s staging, which is essential for an adequate therapeutical response. Accurate staging based on information on intra- and extraluminal pathologic lesions allows adequate therapeutic response.

Pooler et al. discussed the variation in diagnostic performance among radiologists at screening CT colonography [18]. They demonstrated consistency among the radiologists in performing CT colonography screening with histologically confirmed advanced neoplasia in 3.6%; 19.5% of polyps proved to be advanced neoplasia and there was a per-polyp endoscopic confirmation of 93.5% (ranging from 80.0 to 97.6%) [18].

In some cases, CT colonography detects and corrects errors in determining the localization of carcinoma with FCS. FCS is limited in the detection of extraluminal formations, as well as in the detection of diverticula. In these cases, CT colonography with intravenous contrast is recommended. The method accurately indicates the presence of extraluminal obstruction, as it presents in detail the involvement of the intestinal wall. All the cases of extraluminal obturations can be detected only with CTC, and such cases are not rare. In cases of extraluminal tumors, classic CT can give information about the location of the formation, but cannot deliver information about the engagement of the process in the surrounding intestinal loops as Luz et al. demonstrated [17].

Additionally, Pickhardt et al. performed an investigation to determine the rate and type of polyps via CT colonography after initial negative CT colonography screening [19]. They confirmed that the CT colonography screening was low compared to the initial CT screening (3.7% vs. 5.2%) [19].

In cases of obstruction due to non-tumor pathologies, FCS is incomplete and cannot give information proximal to the obstruction because of the edema and fibrotic development. CTC is the only diagnostic method that can reveal the nature of the pathology as well as the engagement of the surrounding tissues. In cases with intraluminal obturation from a tumor formation, although incomplete, FCS establishes the exact cause and material for histological verification because of the disadvantage of FCS related to the lack of information proximal to the obturation area of the intestinal loops. CT colonography using air for colon distension allows a complete check-up of the colon, and the diagnostic process is combined with a proper staging of the disease.

In line with the aforementioned, Graser et al. compared CT colonography, colonoscopy, sigmoidoscopy, and fecal occult blood tests for the detection of advanced neoplasia [20]. Based on their results, they recommend high-resolution and low-dose CT colonography for colorectal screening. If patients refuse full bowel preparation and colonoscopy, flexible sigmoidoscopy should be preferred over stool blood tests.

Often the functional spasm due to the invasiveness of the procedure cannot be recognized by the FCS examination and this is a reason for patients’ poor acceptance rate of the method. CTC is the alternative diagnostic method that can differentiate organic from functional spasms of the colon, as well as diseases connected to changes in the length of the colon [21,22].

In our patients, the main reason for incomplete or refused FSC are intraluminal obturation from a tumor formation; extraluminal obturation by tumor; functional spasm and stenotic changes not caused by cancer; congenital diseases with changes in the length and lumen of intestinal loops; contraindications for performing FCS, and subjective reasons on the part of the patient. Our optimized CT colonography can overcome the mentioned reasons and becomes the method of choice in cases of complete, contraindicated, or refused FCS, as well as in cases of screening for colorectal carcinoma and polyposis and in cases of postoperative control. This is in accordance with the investigations of Atkin et al. [23]. They compared rates of additional colonic investigation after CT colonography or colonoscopy for detection of colorectal cancer/large polyps (>10 mm) in symptomatic patients. The overall conclusion was that CT colonography performed with similar sensitivity but less invasiveness than colonoscopy, therefore, there is a need for guidelines to reduce the referral rate after CT colonography [23].

A multicenter randomized trial SIGGAR confirmed that CT colonography outweighed barium enema for diagnosing colorectal carcinoma, thus, the former should be preferred [24].

Our study was the first in our country confirming that CT colonography is a method of choice in cases of negative FCS results accompanied by clinical data for the neoplastic process and in cases of incomplete and contraindicated FCS. Furthermore, a novelty of our study is optimizing the CT colonography with automatic air insufflation leading to improved quality of the obtained images and the patient’s tolerance to the study. Overall, we established that the main advantage of CT colonography was the lack of sedation, which increases the cost-effectiveness of the method.

To summarize the advantages and disadvantages of virtual colonoscopy, we present them in Figure 8.

## 5. Conclusions

Our results confirmed that CT colonography is a method of choice in cases of negative FCS results accompanied by clinical data for the neoplastic process and in cases of incomplete and contraindicated FCS. Optimized CT colonography with automatic air insufflation improves the quality of the obtained images and the patient’s tolerance to the study. CT colonography is helpful in tumor staging, dynamic control of the treatment, and in cases of congenital diseases leading to abnormal changes in the length of the colon and its diameter (megacolon, dolichocolon). The main advantage of CT colonography is the absence of sedation, which is not only cost-effective but also very important in times of pandemic, allowing maintenance of social distance.

## Figures and Tables

**Figure 1 life-13-01906-f001:**
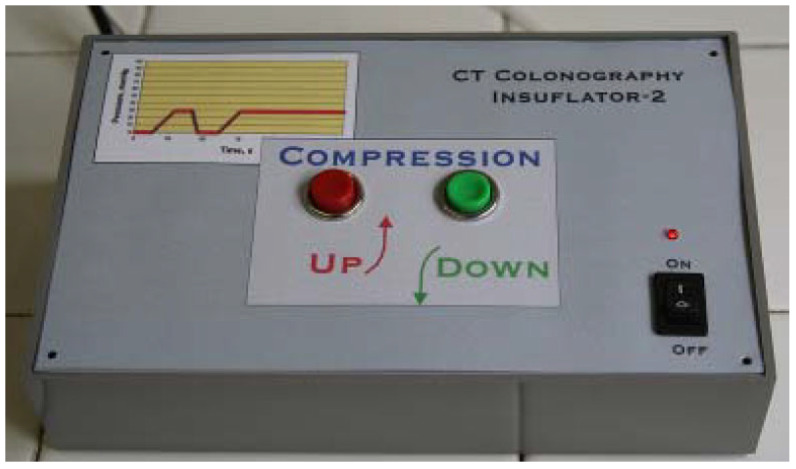
System for automated insufflation of air for CT colonography.

**Figure 2 life-13-01906-f002:**
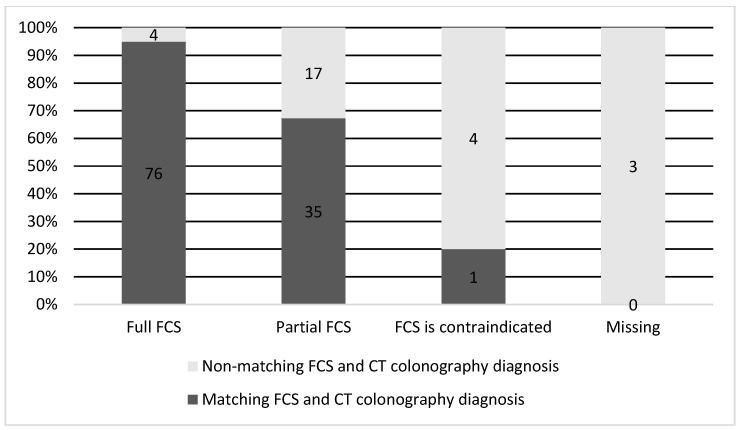
Concordance between the diagnosis made on CT colonography and FCS.

**Figure 3 life-13-01906-f003:**
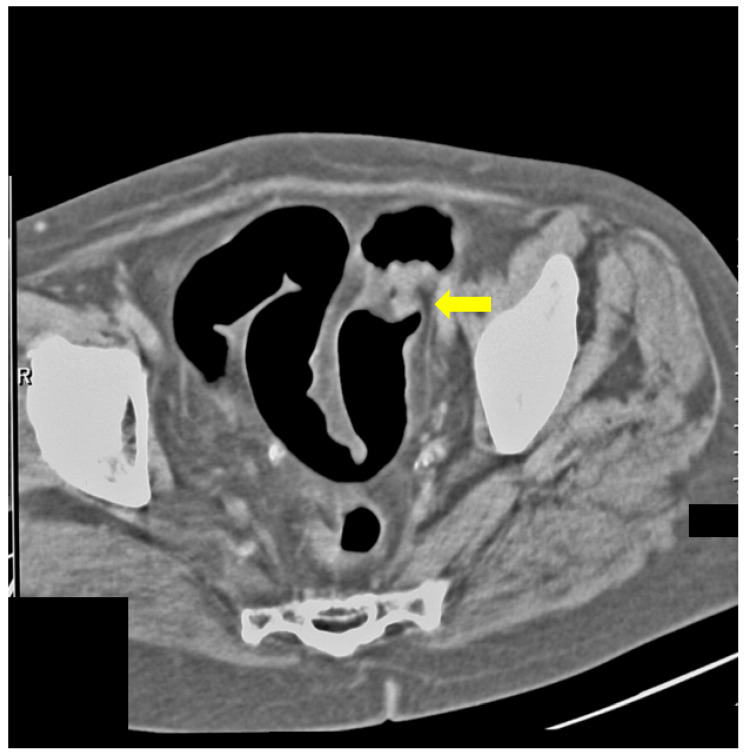
CT colonography-axial scans in a supine position; 73-year-old patient with incomplete FCS due to intraluminal obturation of the sigmoid colon from carcinoma, which is presented in the figure as a tumor with soft tissue density (yellow arrow).

**Figure 4 life-13-01906-f004:**
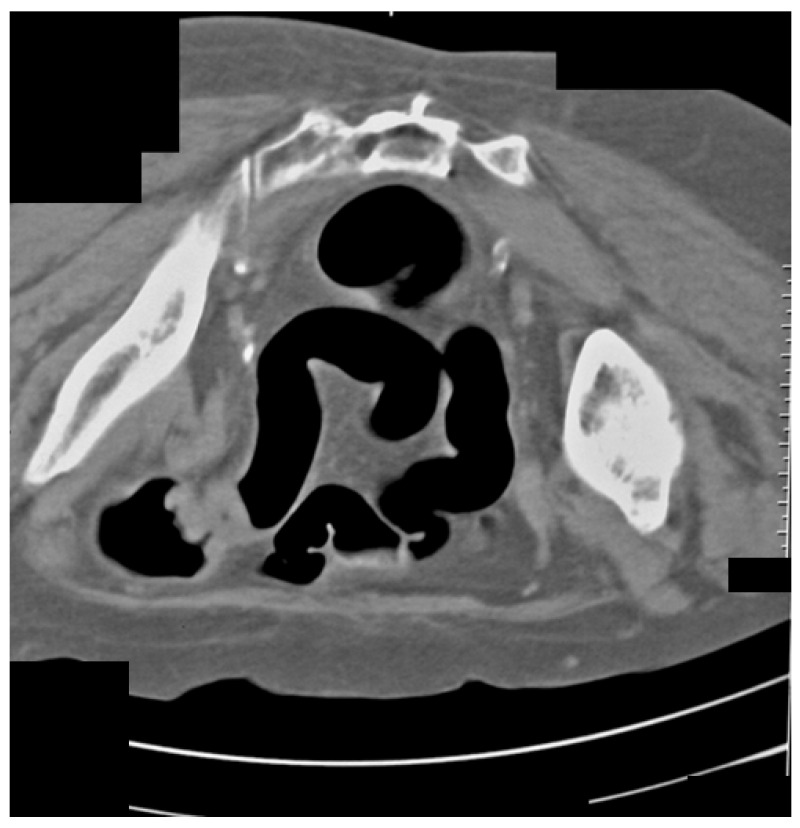
CT colonography-axial scans in a prone position. The same 73-year-old patient with incomplete FCS due to intraluminal obturation of the sigmoid colon from carcinoma is presented in the figure showing a tumor with soft tissue density.

**Figure 5 life-13-01906-f005:**
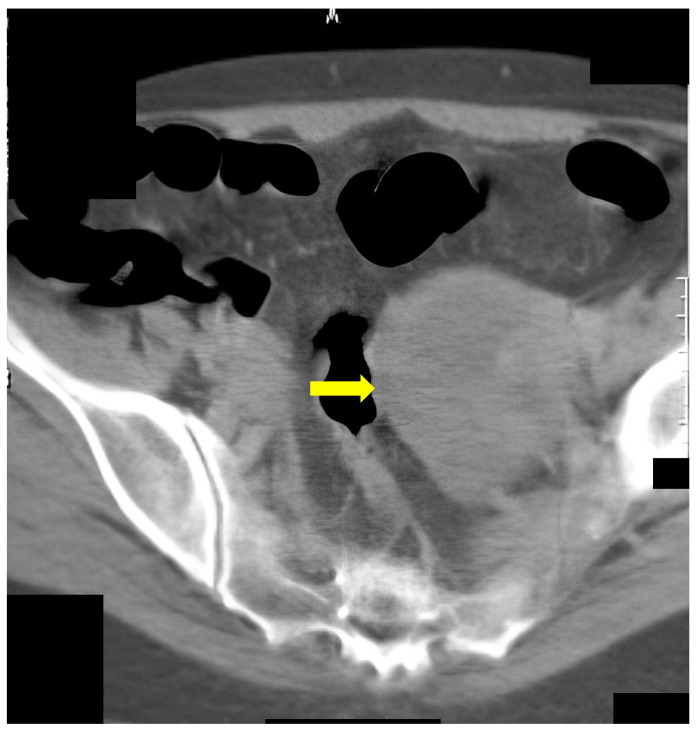
CT colonography–axial slice in a supine position. Narrowing of the sigmoid colon due to a large ovarian tumor in the left is presented (yellow arrow). FCS did not detect any abnormalities.

**Figure 6 life-13-01906-f006:**
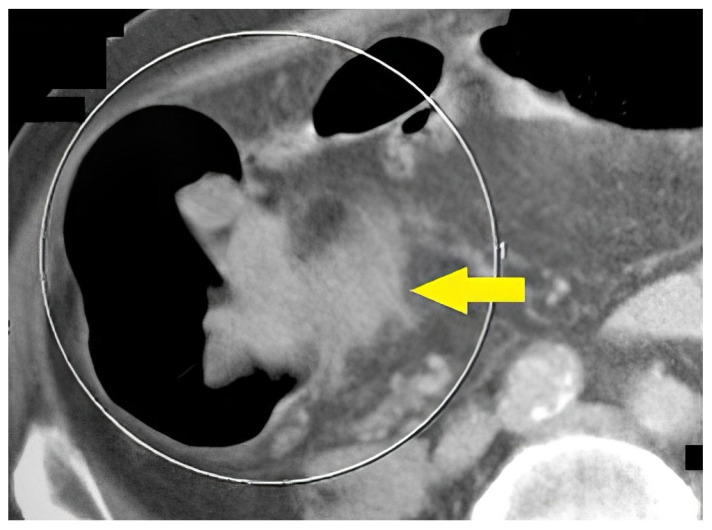
CT colonography-axial slice; 80-year-old patient with incomplete FCS obturation from extraluminal origin was suspected (yellow arrow). After CT colonography, cancer of the cecal valve was found.

**Figure 7 life-13-01906-f007:**
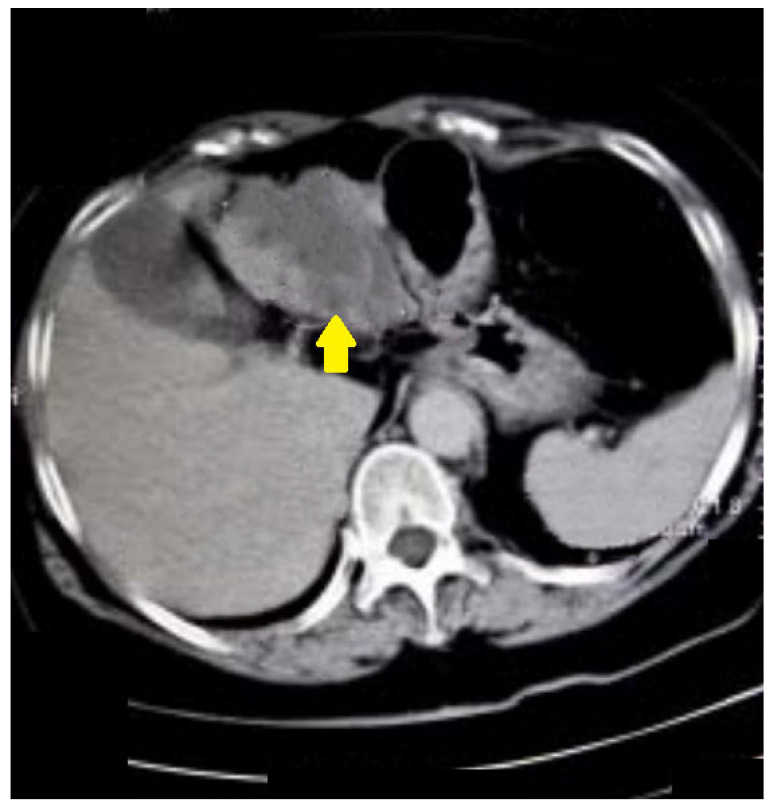
A 67-year-old patient. CT scan, axial view. Incomplete FCS with suspicion of functional spasm. Normal intraluminal view on FCS. After CT colonography, a stomach carcinoma with liver metastases was found (yellow arrow).

**Figure 8 life-13-01906-f008:**
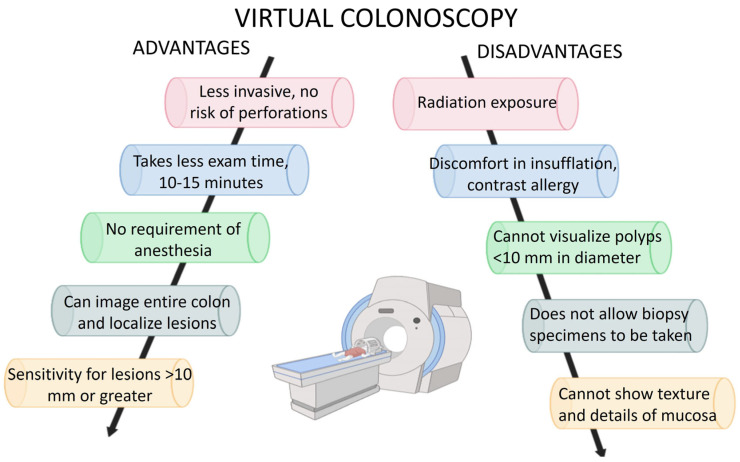
Advantages and disadvantages of CT colonography.

**Table 1 life-13-01906-t001:** Distribution of patients according to the type of antispasmodic administered.

Medicament	Buscolisine	Drotaverine
Number	102	38

**Table 2 life-13-01906-t002:** Distribution of patients according to the type of preliminary preparation.

Medicament	X-Prep	Fortrans	No Prepatation
Percentage	93.60%	3.60%	2.90%

**Table 3 life-13-01906-t003:** Causes for incomplete colonoscopy.

Cause	Patients, Number (%)	Details
Intraluminal obturation of tumor nature	17 (33%)	Usually, these tumors require surgery and chemotherapy/radiotherapy.
Extraluminal obturation (compression) from a tumor formation	4 (8%)	These patients considered with tumor formation from the colon after incomplete colonoscopy to be correctly diagnosed with tumors of the stomach or ovaries.
Stenotic changes of a non-tumor nature	11 (21%)	Stenotic changes of a non-tumor nature include cases of Crohn’s disease and mucosal edema. The reasons for incomplete FCS are entirely inconsistent with the results of CT colonography.
Congenital diseases with changes in the length of the lumen of the intestinal loops (dolichocolon, megacolon, Hirschsprung)	7 (13%)	In 7 patients, the presence of dolicholone with exhaustion of the device was assumed to be the cause of incomplete FCS. CT colonography was normal in 4 of them, and in 3 patients, CT colonography established carcinoma of the cecum and carcinoma of the rectum, respectively.
Subjective factors	13 (25%)	Pain, poor preparation, contraindications.

## Data Availability

Not applicable.

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
