# Peer review of "High Concordance of CT Colonography and Colonoscopy Allows for the Distinguishing and Diagnosing of Intestinal Diseases"

_life, 2023, doi:10.3390/life13091906_

Round 1
Reviewer 1 Report
Very interesting work. Just notice that figures 3 and 8 are not centered in the manuscript file
Author Response
Very interesting work. Just notice that figures 3 and 8 are not centered in the manuscript file
- Thank you for your time to review our paper and for the overall evaluation of our paper as good.
- We acknowledge that our paper might have some issues in conformity with the referee`s comment on figure 3 and 8. We corrected the issue by centering the figures.
Reviewer 2 Report
In this study 140 patients were reviewed with CT 26 colonography, where incomplete colonoscopy was observed in approximately one-third of them.
Fig 2 please consider edit this graphic,Please use black color instead grey.
Fig 3 please use color marker to show clinical result
Fig 5-7 please use color marker to show clinical result
Pleas discuss changes between Figures 3, 4, 5, 6 and 7
Please underline novelty in the introduction and conclusion.
PLease re-do figure 8
Please in discusion section make sure to use current references.
Author Response
In this study 140 patients were reviewed with CT 26 colonography, where incomplete colonoscopy was observed in approximately one-third of them.
- Thank you for your time to review our paper. We acknowledge that our paper might have some issues in conformity with the referees` comments. We have addressed them and revised the manuscript accordingly and to improve the introduction, cited references, design, methods, results and conclusions.
- Changes are visible in red and/or track changes. We believe that you find these changes satisfactory, and the revisions have substantially improved the quality of the manuscript.
Fig 2 please consider edit this graphic,Please use black color instead grey.
- Thank you for the valuable suggestion. We changed the colors in gray scale to be visible even printed in white and black.
Fig 3 please use color marker to show clinical result
- Thank you for the recommendation. We added a yellow arrow to show the important changes in the figure.
Fig 5-7 please use color marker to show clinical result
- Thank you for the recommendation. We added a yellow arrow to show the important changes in the figure.
Pleas discuss changes between Figures 3, 4, 5, 6 and 7
- Thank you for the note. Figures represent images of different patients who are discussed in the text. If the referee could clarify the requirement we could add more information.
Please underline novelty in the introduction and conclusion.
- We accept the critical note. We emphasized the novelty in the introduction and discussion.
PLease re-do figure 8
- Thank you for the note. We revised the figure to make it more clear, with adjusted font, size and colors.
Please in discusion section make sure to use current references.
- Thank you for the critical note. We have added more references from the recent years in the discussion and introduction.
Round 2
Reviewer 2 Report
Thank you